# Poison or Potion: Effects of Sunflower Phenolamides on Bumble Bees and Their Gut Parasite

**DOI:** 10.3390/biology11040545

**Published:** 2022-04-01

**Authors:** Antoine Gekière, Irène Semay, Maxence Gérard, Denis Michez, Pascal Gerbaux, Maryse Vanderplanck

**Affiliations:** 1Laboratoire de Zoologie, Research Institute for Biosciences, University of Mons, 7000 Mons, Belgium; denis.michez@umons.ac.be; 2Organic Synthesis and Mass Spectrometry Laboratory, Research Institute for Biosciences, University of Mons, 7000 Mons, Belgium; irene.semay@umons.ac.be (I.S.); pascal.gerbaux@umons.ac.be (P.G.); 3Insect Lab., Division of Functional Morphology, Department of Zoology, Stockholm University, 11418 Stockholm, Sweden; maxence.gerard@zoologi.su.se; 4CEFE, Univ Montpellier, CNRS, EPHE, IRD, 34293 Montpellier, France

**Keywords:** *Crithidia bombi*, *Bombus terrestris*, *Helianthus annuus*, specialised metabolites, microcolony performance, phenotypic variation, immunocompetence

## Abstract

**Simple Summary:**

Bee declines have been reported worldwide, partly due to parasite spread induced by human activities. However, bees may forage on specific floral resources to face parasite infection. Such natural resources are comparable to ‘natural pharmacies’ and may be favoured in bee conservation strategies. Consumption of sunflower pollen, despite being detrimental for larval development, has been recently shown to reduce the load of a widespread bumble bee gut parasite in the common eastern bumble bee. Although the underlying mechanisms remain unknown, it has been suggested that sunflower phenolamides—a family of molecules found in most flowering plants—may be responsible for such a reduction in parasite load. Here, we tested the impacts of sunflower phenolamides on healthy and infected buff-tailed bumble bees. Expectedly, sunflower pollen had harmful consequences on bumble bee development but surprisingly, it did not alter parasite load. By contrast, sunflower phenolamides had milder effects on bumble bee development but unexpectedly increased parasite load. Phenolamide effects may stem from the physiological stress they induced or the gut microbial community alteration they may have triggered. Since biological models and experimental framework differ greatly in related studies tackling plant–bee–parasite interplays, we challenged the definition of medicinal effects and questioned the way to assess them in controlled conditions.

**Abstract:**

Specific floral resources may help bees to face environmental challenges such as parasite infection, as recently shown for sunflower pollen. Whereas this pollen diet is known to be unsuitable for the larval development of bumble bees, it has been shown to reduce the load of a trypanosomatid parasite (*Crithidia bombi*) in the bumble bee gut. Recent studies suggested it could be due to phenolamides, a group of compounds commonly found in flowering plants. We, therefore, decided to assess separately the impacts of sunflower pollen and its phenolamides on a bumble bee and its gut parasite. We fed *Crithidia*-infected and -uninfected microcolonies of *Bombus terrestris* either with a diet of willow pollen (control), a diet of sunflower pollen (natural diet) or a diet of willow pollen supplemented with sunflower phenolamides (supplemented diet). We measured several parameters at both microcolony (i.e., food collection, parasite load, brood development and stress responses) and individual (i.e., fat body content and phenotypic variation) levels. As expected, the natural diet had detrimental effects on bumble bees but surprisingly, we did not observe any reduction in parasite load, probably because of bee species-specific outcomes. The supplemented diet also induced detrimental effects but by contrast to our a priori hypothesis, it led to an increase in parasite load in infected microcolonies. We hypothesised that it could be due to physiological distress or gut microbiota alteration induced by phenolamide bioactivities. We further challenged the definition of medicinal effects and questioned the way to assess them in controlled conditions, underlining the necessity to clearly define the experimental framework in this research field.

## 1. Introduction

Bees rely on floral resources, mainly pollen, to meet the nutritional requirements for their development, reproduction and survival [1]. Pollen consists of both central (i.e., phytochemicals involved in plant growth and development such as proteins, amino acids, lipids and carbohydrates) [2] and specialised metabolites (i.e., phytochemicals involved in plant abiotic and biotic interactions such as alkaloids, phenolics and terpenoids) [3]. Whereas bee–plant interactions are often regarded as a ‘perfect mutualism’, they actually hide a silent conflict in which bees act concurrently as pollinators, essential to plant reproduction, and palynivores, compromising the plant reproductive success (i.e., pollen dilemma [4]). This conflict of interest has been proposed as an explanation for the occurrence of specialised metabolites in pollen that may be toxic or deterrent to some bee species, which limits excessive pollen harvesting (e.g., [5,6]). Indeed, some specialised metabolites are detrimental to unspecialised bees by impeding larval development [7], leading to gut damage in adults [5], inducing malaise behaviours [8], weakening the immune system [9] and, in some instance, killing larvae and adults [8]. Moreover, other hypotheses have been put forward to explain the presence of specialised metabolites in pollen, such as pleiotropy (i.e., non-adaptative genetic or physiological leakage from other tissues) and protection against biotic (e.g., pathogens) or abiotic (e.g., UV radiation) stressors [10,11]. Along with pollen nutrients, it is likely that pollen-specialised metabolites shape bee–plant interactions.

Bees, even generalist species, do not forage randomly on all available plant species but rather display a selective foraging behaviour, favouring floral resources that meet their nutritional and physiological requirements [12]. Actually, feeding on suitable resource may help bees face environmental stressors such as heat stress [13], pesticide exposure [14] but also parasite infection [15,16]. Indeed, wild bees are challenged with a vast range of parasites, parasitoids and pathogens including metazoans, protozoans, bacteria and viruses [17,18]. Infection outcomes greatly differ regarding enemy species: while some bees’ enemies were found to only have sublethal effects (e.g., impaired flower handling and foraging behaviour [19]), some significantly reduced their host survival (e.g., [20]). Further, outcomes vary depending on the host species and its physiology, including its nutritional state [21,22].

How specific pollen help bees to deal with an infection remains mostly unknown, but studies have suggested that the consumption of particular resources can either impede parasite development (e.g., flagellum removal [23]) or boost bee immunity (e.g., greater fat body content and pro-phenoloxidase production [24]). Notably, the consumption of specific pollen-specialised metabolites has been shown to reduce parasite load in bumble bees [25] and help honeybees to face infection [26]. For instance, sunflower pollen (*Helianthus annuus*; Asterids: Asteraceae) has been shown to reduce *Crithidia bombi* (Euglenozoa: Trypanosomatidae) infection in the common eastern bumble bee *Bombus impatiens* (Hymenoptera: Apidae) [27]. This observation was consistent among several sunflower cultivars and populations [28] but differed according to the timing and duration of exposure to sunflower pollen [29] as well as to the caste of the infected bumble bees [30]. Sunflower pollen has, therefore, been suggested as a suitable resource for infected bumble bees despite its low nutritional quality [31]. Yet, the mechanism underlying such a parasite load reduction remains unknown [32], although it has been recently suggested that it could be due to a more rapid excretion induced by sunflower pollen consumption, and more especially by its phenolamides [33], a major class of phenylpropanoid metabolites evolutionarily conserved across angiosperms [34,35].

Here, we assessed the impacts of sunflower pollen and its phenolamides on healthy and *Crithidia*-infected buff-tailed bumble bees *Bombus terrestris* at the microcolony and individual levels using a fully crossed experimental design (Figure 1; see Appendix B for details about the biological models). More especially, we aimed to determine whether (i) an adaptative allocation of phenolamides occurs among sunflower tissues (i.e., qualitative or quantitative difference among vegetative tissues and floral resources); (ii) sunflower pollen, and particularly its phenolamides, shape interactions with bumble bees through detrimental effects on microcolonies (i.e., development and stress response) or individuals (i.e., immunocompetence and phenotype); and (iii) sunflower pollen, and particularly its phenolamides, display medicinal effects by reducing parasite load or alleviating the infection costs in bumble bees. We found significant differences in phenolamide compositions among sunflower tissues as well as significant impacts of the different diets on healthy and *Crithidia*-infected bumble bees. We also found a significant impact of phenolamides on *Crithidia* load.

## 2. Materials and Methods

### 2.1. Phenolamide Profiling in Sunflower

Pollen, nectar, corolla and leaves were sampled from five sunflower specimens (seeds provided by Ecoflora; Halle, Belgium) within the same location (Bee Garden, UMons; Mons, Belgium) during August 2019 to take into account biological variation among individuals without changing the abiotic conditions (e.g., soil composition, light exposure). On each individual, one or two inflorescences were covered by a net (polyethylene mesh 800 × 1000 μm) to exclude insect visits and left for three days to allow massive pollen and nectar production. In the morning, nectar was collected from each inflorescence by using microcapillaries and pollen by touching the flower with a vibrating tip. Samples were then stored at −20 °C until chemical analyses. This sampling session was non-invasive, preventing chemical modification that may occur in response to plant damage. Afterwards, leaves and corolla were quickly sampled on each individual, placed in aluminium foil and immediately frozen in liquid nitrogen to avoid any biases due to activation of defensive metabolic pathways. Vegetative samples were then stored at −80 °C until lyophilisation (CHRIST^®^ Alpha 1-2LDplus). Lyophilised samples were ground and kept at room temperature in a dark and dry place until chemical analyses. Phenolamide profiling and quantification in the sampled sunflower tissues and resources were conducted using HPLC-MS/MS, following a methanol/water extraction [36,37]. For quantification, synthetised *N,N′,N″*-triferuloyl spermidine was used as internal standard (see Appendix A for details about chemical analyses).

All statistical analyses were run in R version 4.0.3 [38] and plots were made using the R-packages cowplot [39] and ggplot2 [40]. We compared the total phenolamide content between tissues using a non-parametric Kruskal–Wallis test (‘kruskal.test’ command, R-package stats [41]). Because the test returned significant results, we further conducted multiple pairwise comparisons with Bonferroni’s correction to avoid increases in type error I due to multiple testing (‘pairwise.wilcox.test’ command, R-package stats [41]). Differences in phenolamide profiles (i.e., relative abundances expressed as percent of total phenolamide content) among the different plant organs were visually assessed using a principal component analysis (‘PCA’ command, R-package FactoMineR [42]). To test whether phenolamide profiles significantly differed among plants, we ran a permutational multivariate analysis of variance (perMANOVA) using the Euclidean distance and 9999 permutations (‘adonis’ command, R-package vegan [43]). When perMANOVA analyses were significant (*p* < 0.05), multiple pairwise comparisons were conducted between tissue profiles to precisely detect the differences (‘pairwise.adonis’ command [44]) with *p*-values adjustment (Bonferroni’s correction). An indicator species analysis was also conducted to determine whether some phenolamides were indicative of a plant tissue (*p*-values adjustment using Holm’s correction; ‘indval’ command, R-package labdsv [45]).

### 2.2. Bioassays

#### 2.2.1. Experimental Design

The way that sunflower pollen and its phenolamides can impact healthy and infected *B. terrestris* performance as well as *C. bombi* load was investigated in a fully-crossed experimental design using *B. terrestris* microcolonies distributed among six different treatments: (i) uninfected microcolonies fed with a control diet of willow pollen (i.e., *Salix* sp.); (ii) infected microcolonies fed with a control diet of willow pollen; (iii) uninfected microcolonies fed with a natural diet of sunflower pollen; (iv) infected microcolonies fed with a natural diet of sunflower pollen; (v) uninfected microcolonies fed with phenolamide-supplemented willow pollen (i.e., phenolamides from sunflower pollen added to the control diet); and (vi) infected microcolonies fed with phenolamide-supplemented willow pollen (i.e., phenolamides from sunflower pollen added to the control diet). Honeybee-collected willow and sunflower pollen loads were purchased from the company ‘Ruchers de Lorraine’ (Nancy, France) and provided by the INRAE (France), respectively, and then separately ground and mixed with 65% sugar solution (*w*/*w*) to obtain consistent ball-shaped candies (see Appendix A for details about diet preparation).

The experiments were conducted at the University of Mons (UMons; Mons, Belgium) from April to June 2021. Fifteen queenless *B. terrestris* microcolonies were established for each treatment using workers from five different colonies provided by Biobest *bvba* (Westerlo, Belgium) that were equally distributed among the treatments to ensure homogeneity of origin. A total of 90 microcolonies were then monitored. Each microcolony consisted of five workers placed in different plastic boxes (10 cm × 16 cm × 16 cm), following a method adapted from [46]. In infected treatments, workers were inoculated individually with *C. bombi* prior to being placed in their microcolonies (see section ‘Parasite inoculation and monitoring’ for details about inoculation). Microcolonies were reared in a dark room (27 ± 1 °C; 60 ± 10% humidity), fed ad libitum with syrup (water:sugar 35:65 *w*/*w*) and pollen, and manipulated under red light to minimise disturbance. A three-day initiation phase was set during which each microcolony was provided with 1 g of willow pollen (pollen:water:syrup, 37.5:18.75:1, *w*/*w*/*w*), enabling microcolonies to initiate their nest on a common pollen diet as well as effective parasite inoculation in infected microcolonies [47]. After this initiation phase, microcolonies were fed for a 35-day period (i.e., experiment phase) with their respective pollen candies that were freshly prepared and renewed every two days (1–3 g depending on the age of the microcolony) to avoid nutrient alteration and drying out during the experiment. Control boxes that did not include bees were implemented and managed in the same way as the other microcolonies to control for evaporation rate in pollen diets and sugar syrup. Workers that died during the experiment were removed, weighed and replaced by new workers (inoculated or not depending on the treatment) originating from the same foundress colony. Pollen and syrup collections were measured every two days by weighing pollen candies and syrup containers before their introduction into the microcolony and after their removal. Dead workers and ejected larvae were checked every other day. At the end of the experiment, workers and emerged males were weighed, the brood was carefully dissected, and the number and mass of individuals were recorded for each developmental stage (i.e., eggs, non-isolated larvae, isolated and pre-defecating larvae, isolated and post-defecating larvae, pupae, non-emerged and emerged males).

#### 2.2.2. Parasite Inoculation and Monitoring

For microcolony inoculation, faeces were collected from 45 workers from three colonies used as parasite reservoirs and from three infected queens collected in natura that were used to implement these parasite reservoirs (see Appendix A for details about the implementation of parasite reservoirs). The use of faeces from different colonies allows to obtain multiple-strain inocula and minimises the risk of specific genotype–genotype interactions since it is likely that different *C. bombi* strains have developed in each infected colony and queen [48]. Faeces were pooled, diluted with 0.9% NaCl solution to make a 1 mL solution and purified following a ‘triangulation’ method developed by [49] and adapted by [50]. A few microliters of the resulting solution were placed in a Neubauer chamber for *C. bombi* cell concentration to be measured. The resulting solution was adjusted to 2500 cells µL^−1^ with 40% sugar solution (*w*/*w*). Workers assigned to infected treatments were isolated in individual Nicot cages, starved for 5 h, and then fed with a 10 µL drop of inoculum containing 25,000 cells, which lies within the range of *C. bombi* cells shed by infected workers [47]. Only workers that consumed the whole inoculum were considered for microcolony establishment.

The first measure was taken three days after inoculation and following measures were then taken every three days. Workers were individually placed in 50 mL Falcon tubes, faeces were collected using a 10 µL microcapillary and pooled by microcolony (i.e., 15 faecal samples per treatment). Each microcolony faecal sample was then diluted (i.e., dilution 5× or 10× according to the load) to allow for counting the *C. bombi* cells by using an improved Neubauer haemocytometer at 400-fold magnification under an inverted phase contrast microscope (Eclipse Ts2R, Nikon; Tokyo, Japan). Workers placed in the microcolony over the course of the experiment for replacing dead ones were not sampled to avoid any bias in parasite load assessment.

Difference in parasite load among treatments was assessed through a generalised linear mixed model (GLMM) with negative binomial errors (log link) using pollen diet and day as fixed effects, and microcolony nested within colony as random effect to account for repeated measures. The model was fitted using maximum likelihood estimation via Template Model Builder (‘glmmTMB’ function, R-package glmmTMB [51]).

For all GLMMs, overdispersion and zero-inflation were checked using the functions ‘testDispersion’ and ‘testZeroInflation’, respectively (R-package DHARMa [52]). We selected the GLMMs using an information-theoric approach based on Akaike’s information criterion corrected for small sample sizes (AICc; ‘model.sel’ command, R-package MuMIn [53]). For each response variable, we compared a set of candidate models, including a full model, all biologically meaningful subsets of the full model and a null model only containing the intercept and random effect. Akaike’s weight was used to choose the best fitting model, with model averaging where no single model had ≥95% AIC support (‘model.avg’ command, R-package MuMIn [53]). The candidate set of models was chosen by adding the next best supported model until a cumulative ≥ 95% support was reached. Regarding statistical significance, we decided to use confidence intervals (CI) rather than the conventional hypothesis testing and a predetermined *p*-value [54]. Parameter estimates (PE), standard errors (SE) and CI were based on full-set averaging of the 95% confidence set. Confidence intervals not crossing zero indicated a significant effect (see Appendix A for AICc, ΔAICc and AICc weight associated to each model as well as Appendix A for PE, SE and 95% CI derived from model averaging).

#### 2.2.3. Microcolony Parameters

Microcolony development and feeding response were evaluated based on (i) pollen and (ii) syrup collection as well as on (iii) the number and mass of individuals for each developmental stage (i.e., eggs, non-isolated larvae, isolated and pre-defecating larvae, isolated and post-defecating larvae, pupae, non-emerged and emerged males) within the brood. Microcolony stress responses were assessed via (iv) larval ejection (i.e., number of larvae removed from the brood by workers over the experiment divided by the number of hatched offspring), (v) pollen efficacy (i.e., the mass of alive hatched offspring divided by total mass of collected pollen), (vi) pollen dilution (i.e., total mass of collected syrup divided by the total mass of collected pollen) and (vii) worker mortality (i.e., number of dead workers over the experiment) [55]. All mass parameters (i.e., brood mass, pollen and syrup collection) were standardised by the total mass of workers in the microcolony to avoid potential bias from worker activities.

Differences in resource collection (pollen and syrup), microcolony development (total offspring mass and number of individuals within each developmental stage) and stress responses (larval ejection, pollen efficacy and pollen dilution) were assessed through GLMMs with pollen diet, infection status and their interaction as fixed effects as well as colony of origin as a random effect (‘glmer’ command, R-package lme4 [56]). The models fitted for pollen and syrup collection used a Gamma distribution and a log link. They also included day and its interactions as fixed effects and microcolony was nested within colony as a random effect to account for repeated measures. Total mass of alive hatched offspring, pollen efficacy and pollen dilution were also analysed using models with a Gamma distribution and a log link. Models assessing the number of individuals within each developmental stage used a Poisson distribution and a log link. Larval ejection was analysed using a binomial model and a logit link with the number of ejected larvae and the number of living hatched offspring produced per microcolony as a bivariate response. When overdispersion occurred in models with a Poisson distribution, an observation-level random effect was added to the model (i.e., microcolony nested within colony as random effect). If overdispersion still occurred, GLMMs with negative binomial errors (log link) were fitted using maximum likelihood estimation via Template Model Builder (observation-level random effects were not considered in these models). Mixed effects Cox proportional hazards models (‘coxme’ function, R-package coxme [57]) were used to analyse mortality risk of workers over the experiment using pollen diet, infection status and their interaction as fixed effects, and colony of origin as random effect. Proportionality of hazards was checked to validate the Cox proportional hazards assumption (‘cox.zph’ function, R-package survival [58]).

For all GLMMs conducted on microcolony parameters, we used the same statistical procedure as previously described (i.e., detection of overdispersion and zero-inflation, model selection based on AICc, use of CI for assessment of statistical significance).

#### 2.2.4. Individual Parameters

As an indicator of immunocompetence [59], abdominal fat body content of 30 workers and 30 emerged males per treatment (i.e., two workers and two males per microcolony) was measured at the end of the bioassays using Ellers’ procedure [60,61]. Briefly, isolated abdomens were weighed before and after drying at 70 °C for three days. They were then placed into 2 mL of diethyl ether for 24 h to extract fat, rinsed twice, and weighed again after drying at 70 °C for seven days. The fat mass proportion of an individual’s abdomen was defined as the abdominal mass loss during this process divided by the individual’s abdomen mass before extraction.

Differences in fat body content between treatments were examined using a GLMM with a Gamma distribution and a logit link to deal with proportion data. Fixed effects included pollen diet, infection status, caste and their interactions while colony of origin was assigned as a random effect.

At the end of the bioassays, analyses of wing size—a proxy for the canalisation of the phenotype—and fluctuating asymmetry (FA)—a proxy for developmental stability—were conducted following [62]. Our total dataset contained 250 males (i.e., 50 males per treatment). The right and left forewings of each specimen were removed, placed on a glass slide and photographed (n = 500 pictures) using an Olympus SZH10 microscope with an AF-S NIKKOR 18–105 mm (Shinjuku, Japan) and GWH10X-CD oculars coupled with a Nikon D610 camera (Shinjuku, Japan). Five individuals were discarded because their wings were damaged or considered as outliers because of wing anomalies such as a missing landmark. Pictures were uploaded in the tpsUTIL 1.81 software [63] and digitised with a set of 18 two-dimensional landmarks in the tps DIG 2.31 software [64] (see [62] for landmark positions). Each landmark coordinate was then multiplied by its scale factor provided for each specimen (‘readland.tps’ command, R-package geomorph [65]). While both wings of each individual were used for FA analysis, only the right wing was used for wing size and shape analyses. We used the Generalised Procrustes Analysis superimposition method to remove all non-shape components by translating specimens to the origin, scaling and rotating each landmark configuration to minimise the distance between each corresponding landmark of each landmark configuration (‘gpagen’ command, R-package geomorph [65]). Centroid size (i.e., the square root of the sum of squared distances between all landmarks and their centroid) of the right wings was used as a wing size estimator. We calculated individual vectors of size FA by subtracting the centroid size of the right and left wings of each individual and selecting the absolute value of this subtraction. We then assessed individual vectors of shape FA by calculating the square root of the sum of each squared value of each landmark (x and y).

Centroid size as well as size and shape FA were assessed through GLMMs with a Gamma distribution and a log link, considering pollen diet, infection status and their interaction as fixed effects, and colony of origin as a random effect. For size and shape FA, we ran the analyses with and without outliers (i.e., removing values above Q3 + 3*IQR or below Q1 − 3*IQR) and reported results under both conditions. Further, a linear model with randomised residuals in a permutation procedure was fitted to understand the effects of pollen diet, infection status and their interaction on emerged male right forewing shape (‘lm.rrpp’ command, R-package RRPP [66])—using an ordinary least squares (OLS) estimation of coefficients on multidimensional data and a randomised residual permutation method, with colony of origin as random effect. Then, a type-II analysis-of-variance table was computed and F-tests were calculated on the full model to assess the significance of explanatory variables.

For all GLMMs conducted on individual parameters, we used the same statistical procedure as previously described (i.e., model selection based on AICc, use of CI for assessment of statistical significance).

## 3. Results

### 3.1. Phenolamides in Sunflower

Total phenolamide content varied among tissues (χ^2^ = 16.573, df = 3, *p* < 0.001), with leaves and corolla displaying a significantly lower content than pollen and nectar, which did not differ from each other (Table 1). Five distinct phenolamide compounds were detected in sunflower floral resources, with different phenolamide profiles between pollen and nectar (F_3,16_ = 33.158, *p* < 0.001, Appendix A). Pairwise comparisons arranged the tissues in three groups: (i) one group with leaf and corolla, without any detected phenolamide, (ii) one with nectar, and (iii) one with pollen (Appendix A). While all detected phenolamides were indicative of floral resources, *N,N′*-diferuloyl spermidine was indicative of pollen (*p* = 0.035, indicator value = 1), and *N,N′,N″,N‴*-tetracoumaroyl spermine (*p* = 0.035, indicator value = 0.712) and *N,N′,N″*-tricoumaroyl spermidine (*p* = 0.035, indicator value = 0.675) were indicative of nectar (Appendix A).

### 3.2. Microcolony Performance

#### 3.2.1. Resource Collection

Microcolonies fed the supplemented diet collected significantly less pollen than microcolonies fed the control diet (supplemented diet: PE = −0.406, CI = −0.607 to −0.206) while pollen collection did not differ from control in microcolonies fed a natural diet (natural diet: PE = −0.045, CI = −0.249 to 0.158). Pollen collection increased over time (day: PE = 0.073, CI = 0.061 to 0.084) but to a lesser extent in microcolonies fed a natural diet compared to those fed a control diet (natural diet*day: PE = −0.064, CI = −0.080 to −0.047) (Figure 2A). Regarding syrup collection, microcolonies fed a natural diet collected significantly more syrup than those fed a control diet (natural diet: PE = 0.222, CI = 0.103 to 0.341) while no difference occurred between microcolonies fed a supplemented diet and those fed a control diet (supplemented diet: PE = 0.088, CI = −0.030 to 0.206). As for pollen, syrup collection significantly increased over time (day; PE = 0.028, CI = 0.025 to 0.030) but to a greater extent in microcolonies fed a control diet (natural diet*day: PE = −0.028, CI = −0.030 to −0.025; supplemented diet*day: PE = −0.009, CI = −0.012 to −0.006) (Figure 2B). Parasite infection never impacted resource collection, regardless of the diet (factor not retained in the final model, see Appendix A).

#### 3.2.2. Stress Responses

Microcolonies fed a natural diet displayed a significantly greater larval ejection than those fed a control diet (natural diet: PE = 1.223, CI = 0.658 to 1.789) while microcolonies fed a supplemented diet did not show any significant difference compared to those fed a control diet (supplemented diet: PE = −0.053, CI = −0.597 to 0.490; Figure 3A). By contrast, no diet effect occurred on worker mortality (factor not retained in the final model, see Appendix A). Parasite infection did not have any impact on either larval ejection (parasite: PE = −0.635, CI = −1.459 to 0.188) or worker mortality (factor not retained in the final model, see Appendix A).

Regarding pollen efficacy, it was significantly lower in microcolonies fed a natural diet compared to those fed a control diet (natural diet: PE = −1.328, CI = −1.596 to −1.061) while pollen efficacy did not differ between microcolonies fed supplemented and control diets (supplemented diet: PE = 0.060, CI = −0.180 to 0.300, Figure 3B). Moreover, microcolonies fed natural and supplemented diets displayed a greater pollen dilution than those fed a control diet (natural diet: PE = 0.343, CI = 0.244 to 0.441; supplemented diet: PE = 0.355, CI = 0.257 to 0.454; see Appendix A). Parasite infection did not influence either pollen efficacy (parasite: PE = −0.098, CI = −0.130 to 0.109) or pollen dilution (parasite: PE = 0.010, CI = −0.060 to 0.079).

#### 3.2.3. Development

While no difference occurred in the number of eggs and non-isolated larvae among diet treatments, significant differences were detected in more advanced brood stages. Indeed, microcolonies fed a natural diet had fewer pre- and post-defecating larvae (natural diet: PE = −3.018, CI = −4.338 to −1.699 and PE = −2.744, CI = −3.673 to −1.815, respectively), pupae (natural diet: PE = −2.069, CI = −2.968 to −1.170) and emerged males (natural diet: PE = −1.699, CI = −2.079 to −1.318) compared to those fed a control diet (Figure 4). In the same way, fewer males emerged in microcolonies fed a supplemented diet compared to those fed a control diet (HCAA: PE = −0.363, CI = −0.600 to −0.125) (Figure 4). Regarding the total mass of alive hatched offspring, it was significantly lower in microcolonies fed natural (natural diet: PE = −1.918, CI = −2.225 to −1.611) and supplemented (supplemented diet: PE = −0.410, CI = −0.700 to −0.120) diets compared to those fed a control diet. Parasite infection did not influence either the number of individuals per developmental stage or the total mass of hatched alive offspring (factor not retained in the final models, see Appendix A).

### 3.3. Infection and Immunocompetence

#### 3.3.1. Parasite Load

Microcolonies fed a supplemented diet had higher infection intensity than microcolonies fed the control diet (supplemented diet: PE = 0.411, CI = 0.109 to 0.774) while no significant difference was observed between microcolonies fed natural and control diets (natural diet: PE = 0.002, CI = −0.350 to 0.355). In all diet treatments, parasite load significantly increased over time (day: PE = 0.064, CI = 0.054 to 0.074) (Figure 5).

#### 3.3.2. Fat Body Content

The fat body content was reduced in both workers and males from microcolonies fed natural and supplemented diets compared to those fed a control diet (natural diet: PE = −0.826, CI = −1.130 to −0.523; supplemented diet: PE = −0.757, CI = −1.060 to −0.455) (Figure 6A). Parasite infection also induced a significant reduction in fat body content (parasite: PE = −0.466, CI = −0.774 to −0.157) (Figure 6B).

### 3.4. Phenotypic Variation

#### 3.4.1. Centroid Size

Centroid size analyses (right wing) indicated that males fed natural and supplemented diets during their development were smaller than those fed a control diet, this effect being more pronounced for the natural diet (natural diet: PE = −0.267, CI = −0.290 to −0.243; supplemented diet: PE = −0.030, CI = −0.053 to −0.007) (Figure 7A). Parasite infection did not significantly impact the centroid size of emerged males although an increasing trend could be observed (parasite, parasite*natural diet, parasite*supplemented diet: PE > 0).

#### 3.4.2. Wing Shape

Regarding the right-wing shape, analyses indicated that males fed a natural diet during their development had greater wing shape dissimilarities than males fed a supplemented diet in comparison to those fed a control diet (natural diet: PE = 0.015, CI = 0.011 to −0.020; supplemented diet: PE = 0.008, CI = 0.004 to 0.013). In the same way, parasite infection induced a shape divergence in emerged males but to a lesser extent than between the diet treatments (parasite: PE = 0.004, CI = 0.000 to 0.007).

#### 3.4.3. Fluctuating Asymmetry

Analyses on FA showed that diet impacted both size and shape FA with males fed natural and supplemented diets during their development displaying a greater FA than those fed a control diet (*Size*—natural diet: PE = 0.374, CI = 0.055 to 0.694; supplemented diet: PE = 0.990, CI = 0.655 to 1.324; *Shape*—natural diet: PE = 0.182, CI = 0.098 to 0.266; supplemented diet: PE = 0.272, CI = 0.188 to 0.356). However, only for shape FA did the difference still occur after removing the outliers (natural diet: PE = 0.181, CI = 0.120 to 0.241; supplemented diet: PE = 0.124, CI = 0.063 to 0.185) (Figure 7B). Parasite infection did not influence either size (*Outliers included*—parasite: PE = −0.001, CI = −0.159 to 0.157; *Outliers excluded*—parasite: PE = −0.008, CI = −0.125 to 0.109) or shape FA (*Outliers included*—parasite: PE = 0.001, CI = −0.028 to 0.030; *Outliers excluded*—factor not retained in the final model, see Appendix A).

## 4. Discussion

### 4.1. Phenolamide Allocation in Sunflower

While no detectable phenolamide occurred in sunflower leaves and corolla, floral resources displayed five phenolamide compounds; namely *N,N′,N″*-dicoumaroyl feruloyl spermidine, *N,N′,N″,N‴*-tetracoumaroyl spermine, *N,N′,N″,N‴*-tricoumaroyl feruloyl spermine, *N,N′,N″,N‴*-tricoumaroyl feruloyl spermine and *N,N′*-diferuloyl spermidine, the latter being specific to nectar. These results are only partly in line with previous screenings of phenolamides in sunflower floral resources as we reported some compounds for the first time and did not detect others (i.e., *N,N′*-dicoumaroyl spermidine and putrescine derivatives) [3,32,67]. The similar phenolamide profiles between sunflower pollen and nectar warrant further investigations since we cannot rule out (i) a potential ‘phenolamide leakage’ between these two resources or (ii) a cross-contamination during sampling sessions (see [68] for details about the morphology of sunflower florets). Regarding variability among plant individuals, two explanations may be provided: (i) although sunflower seeds were sowed simultaneously, ages could differ among flowering plants at sampling session and result in phytochemical differences (e.g., [69]); and (ii) inter-cultivar variation [3] cannot be ruled out as we were unaware whether sunflower seeds were of the same variety (Pers. Comm. From Ecoflora; Halle, Belgium).

Although the roles of phenolamides in plant development and resistance against abiotic (e.g., UV radiation) and biotic (e.g., pathogens and herbivores) stressors are well-documented [34], no function has been clearly attributed for their large amount in pollen (coat) and nectar [70]. However, the evidence is that phenolamides in sunflower floral resources do not ‘simply’ result from a pleiotropic effect as none of them were detected in vegetative parts. Moreover, since sunflower nectar is a phloem derivative [68], it suggests that pre-nectar from the vascular system is free of phenolamides, which are then probably synthesised in the nectariferous tissues. The origin of phenolamides in sunflower pollen is not clear as they can arise from both leakage from anthers [71] and biosynthesis in pollen cytoplasm [72]. Regardless, such an occurrence of phenolamides in floral resources obviously exposes bees to their effects, especially since these specialised metabolites are widespread among flowering plants [34].

### 4.2. Effects of Sunflower Pollen and Phenolamides on Bumble Bees

Our bioassays corroborate the well-known unsuitability of sunflower pollen for bumble bees [31,46,73] as microcolonies fed a natural diet displayed a lower pollen efficacy, a greater larval ejection and impeded brood development compared to those fed a control diet. Sunflower pollen is not suitable for bumble bee development mainly due to nutritional deficiencies [74] but also to the occurrence of potentially toxic specialised metabolites (e.g., alkaloids [3]) and to the peculiar morphology of exine, which is typical of Asteraceae [75,76]. As no compensatory feeding behaviour (i.e., increased pollen collection) has been highlighted, it suggests that unsuitability may be due to pollen toxicity or low digestibility rather than to nutritional deficiencies (e.g., [5]). Actually, microcolonies fed a supplemented diet also produced a reduced number of males and a restricted offspring mass, suggesting that phenolamides may partly explain sunflower unsuitability. They also displayed a higher pollen dilution, which is known as a behaviour allowing for mitigation of unfavourable pollen properties (e.g., [55]). However, neither natural nor supplemented diets induced mortality among the *B. terrestris* workers. Such unsuitable but sublethal effects of sunflower phenolamides might arise from their antifungal and antibacterial properties [67,77] that may disrupt the bumble bee microbiota (e.g., by boosting/depleting some phylotypes [78]). Moreover, phenolamides are known to upregulate some genes in bees that are homologous to those that stimulate rapid excretion in other insects [79], suggesting that phenolamides might negatively alter bumble bee physiology.

Alongside these effects reported in the literature, our bioassays highlighted that phenolamides also induced a reduction in fat body content, which is a major component of the immune system [59]. Indeed, individuals fed a supplemented or natural diet displayed lower fat body content compared to those fed a control diet. As natural and supplemented diets have different nutritive composition (i.e., one based on sunflower pollen, the other based on willow pollen as for the control diet), this effect cannot arise from differences in pollen nutrients or digestibility. The only valid explanation would be the contribution of the fat body in the detoxification of allelochemical compounds [80]. Actually, the occurrence of phenolamides in both natural and supplemented diets could have activated detoxification pathways, leading to metabolic costs associated with a reduction in lipid reserves (i.e., fat body).

Another noticeable effect of phenolamides on newly emerged males was the reduced centroid size (i.e., males fed natural and supplemented diets were smaller compared to those fed a control diet), which reflects stressful developmental conditions and affects the selective value of individuals [81]. This observation is in line with the stress responses and the impeded brood development observed in the microcolonies fed natural and supplemented diets compared to those fed a control diet. Such effects of diets on offspring size have been already observed in several studies (e.g., [82]), including one on sunflowers [46]. Regarding phenotypic variation, the occurrence of phenolamides in the pollen diet also impacted the shape of the forewing and increased shape FA in newly emerged males (i.e., significant differences among males reared on different diets). Such effects of specialised metabolites on male wing shape have already been highlighted in similar bioassays using sinigrin and amygdalin-supplemented diets [62] and can be indicative of changes in environmental conditions or presence of stressors [83]. By contrast, levels of FA are often lower under controlled conditions [62], and the increase in FA in stressful conditions is indicative of a lesser developmental stability, meaning that males are challenged during their development, which leads to deviations from perfect symmetry between each side [84]. The mechanism explaining such modifications under various stressors remains unclear, but it has been proposed that a shift in energy allocation (e.g., activation of detoxification pathways) can occur and then weaken the homeostasis, ultimately impacting the phenotype [85].

### 4.3. Infection Costs of a Gut Parasite on Bumble Bees

While diet effects on microcolonies were strongly pronounced, only mild effects of the gut parasite *Crithidia bombi* were observed in our study since infected microcolonies did not display neither higher mortality nor higher stress responses than uninfected ones. Actually, *C. bombi* is known to be a highly prevalent but not too virulent gut parasite [86]. Indeed, previous studies only reported sublethal outcomes (e.g., impeding foraging behaviours and cognitive functions [19]), except under food-limited conditions [21] (but see [87]). While a compensatory feeding behaviour (i.e., increased pollen collection) and reduced survival have been highlighted in infected *B. impatiens* workers [87], these effects were not observed herein for *B. terrestris*, suggesting that differences in susceptibility occur among bumble bee species.

As expected, due to the immune challenge (i.e., lipid mobilisation in the haemolymph [59]), parasite infection resulted in a significant decrease in fat body content (i.e., infected individuals displayed lower fat body content compared to uninfected ones). However, such an effect has never been highlighted in previous laboratory studies investigating the impact of the parasite on bumble bees’ fat bodies, probably because of the experimental design that implied isolating each individual for a short period [21,88]. In our bioassays, individuals were maintained within microcolonies and likely constantly reinfected themselves via nestmate faeces and brood with a continuous exposure for 35 days [89], which could have resulted in a greater immune challenge than in these previous studies.

Regarding the phenotypic variation, the parasite did not impact any of the measured parameters (i.e., centroid size, wing shape, size FA, and shape FA). This result contrasts with a previous study showing that *Apicystis bombi* (Apicomplexa: Neogregarinorida) impacted wing size and shape as well as size FA [62]. This discrepancy could be explained by the difference in parasites’ host life stages. Indeed, *A. bombi* infects all brood stages as well as adults, challenging the bumble bees during their development, whereas *C. bombi* only infects adults without any developmental challenge [89].

### 4.4. Effects of Sunflower Pollen and Phenolamides on a Gut Parasite

Parasite infection monitoring during our bioassays clearly indicated that both natural and supplemented diets did not reduce the parasite load in infected microcolonies compared to the control diet. This observation is quite surprising and unexpected given the plethora of previous studies that systematically found a medicinal effect of sunflower pollen in infected bumble bees in different experimental designs (i.e., different sunflower cultivars, different parasite strains, workers housed individually or in microcolonies) [27,28,29,30,31,32]. One explanation would be the difference in the host bumble bee species as we used *B. (Bombus) terrestris* whereas previous studies used *B. (Pyrobombus) impatiens*, which modifies the host genotype *x* parasite genotype *x* environment interacting factor in the *Bombus*–*Crithidia* system [22]. Such a discrepancy has recently been demonstrated by Fowler et al. [90] who showed that sunflower pollen reduced *Crithidia* load in *B. impatiens*, *B. bimaculatus* and *B. vagans* (subgenus *Pyrobombus*) but not in *B. griseocollis* (subgenus *Cullumanobombus*). Another difference in our experimental design is the delay between parasite inoculation and consumption of sunflower pollen, which seems to be a crucial parameter when assessing the medicinal effect of pollen diet. Indeed, previous studies indicated that infected bumble bees fed sunflower pollen 3.5 days after inoculation did not display any reduced parasite load compared to control, whereas infected bumble bees fed sunflower pollen right after inoculation displayed a reduced parasite load after seven days [29] (but see [27]). As our experimental design included an initiation phase after inoculation (i.e., all microcolonies fed for three days on the control diet), there was a delay prior to the consumption of sunflower pollen that could account for the absence of a medicinal effect. Discrepancies may also arise from difference in pollen used as the control diet, namely willow in our study and buckwheat or wildflower mix in previous ones [27,28,29,30,31,32]. Moreover, because of different methods for assessing parasite load (i.e., count in faeces not to kill individuals vs. count in gut suspension), it was not possible to compare infection intensities with those of previous studies. We are then unable to clearly establish which experimental parameter (i.e., bumble bee species, delay between inoculation and sunflower pollen feeding, control diet) was responsible for the absence of a medicinal effect of sunflower pollen (compared to control pollen) in our bioassays, which would partly bring insight for the underlying mechanisms and therefore warrant further investigations.

While no effect of sunflower pollen has been highlighted for parasite infection, phenolamides benefited the parasite as infected workers fed a supplemented diet displayed an increased parasite load. Although host feeding seems to favour parasite cell growth [47,91], differences in feeding behaviour cannot explain our observations as microcolonies fed a supplemented diet did not display higher pollen collection compared to those fed a control diet. Such a fluctuation in parasite load is then likely related to the occurrence of phenolamides themselves, as already shown for other specialised metabolites [23,87]. Such a parasite-facilitating effect of phenolamides has been already observed in a previous study, though it was less pronounced than herein, probably because of some differences between the experimental designs [32]. This effect could arise from different biological activities of phenolamides: (i) their antifungal and antibacterial properties [67,77] that may disrupt the gut microbiota and then weaken a crucial non-immunological defence [92]; (ii) their antioxidant and radical scavenging activities that may lead to a decrease in reactive oxygen species and an immunosuppressed state [93]; and (iii) their potential toxic activities that could result in activation of defence pathways (i.e., detoxification system), altering bee physiology, consuming their energy reserves and then weakening the whole organism that would not be disposed to face an immune challenge (e.g., [94]). While the latter hypothesis is supported by the significant reduction in fat body content in our bioassays, it cannot be the unique reason for the observed increase in parasite load since natural diet (sunflower pollen) also reduced the bumble bees’ fat bodies without impacting the parasite load. Such a mechanistic process should be partly elucidated by assessing the effects of phenolamides on *C. bombi* growth through in vitro assays (e.g., [95]). While we cannot unravel the mechanisms favouring parasite development in microcolonies fed a supplemented diet, we can however propose that phenolamides are not responsible for medicinal effects of sunflower as recently suggested by [33], that put forward a reduction in parasite load through more rapid excretion after sunflower pollen consumption. Further experiments are therefore required to elucidate the chemical or physical mechanisms underlying the medicinal effects of sunflower pollen.

### 4.5. Focus for Future Research

Based upon the numerous experimental works reviewed above, one can arguably state that assessing the medicinal effects of a specific pollen diet is not an easy task, as it requires choosing the right control diet and clearly defining what could be considered as a medicinal effect. Most of the experimental designs assessing pollen medicinal effects rely on parasite cell counts between different diets, including a control diet. Indeed, the host genotype *x* parasite genotype *x* environment interacting factor renders the absolute value of parasite load senseless, making comparisons across treatments compulsory to yield relative results. However, no diet has been clearly established as a universal control. Since all pollen diets display specialised metabolites with differing biological activities and potential physical properties, no one can clearly rule out potential effects of their control pollen diet on parasite load (i.e., favouring or impeding effects). One solution would be to use an artificial diet free of specialised metabolites and physical barriers, and suitable for microcolony development, or to clearly establish the absence of medicinal effects of the natural control pollen diet prior to bioassays (e.g., examining parasite growth through in vitro assays), which has never been thoroughly done. Besides this issue, the definition of ‘medicinal effects’ itself may be confusing. Importantly, a medicinal effect may occur either by benefiting the host or by hampering parasite growth. While some have claimed that a diet must compulsorily be detrimental to the parasite to be considered as medicinal [96], others have argued that it is not mandatory and proposed that medicinal diets could either increase host resistance or tolerance to infection [97]. Furthermore, detrimental effects on unicellular parasites cannot be only assessed based on cells count but should also be considered through molecular impacts on parasite cells such as impairment of protein synthesis, intercalation in DNA, disruption of cell wall, induction of apoptosis, or any other mechanism impeding parasite fitness such as flagellum loss (e.g., [23]). Experiments seeking the most suitable control diet when addressing medicinal effects of pollen on infected bees as well as experiments testing the molecular effects of pollen-specialised metabolites on parasite cells promise to bring new insights into the mechanisms underlying medicinal effects of pollen. Untangling such mechanisms would shed light on the way bees could use floral resources to overcome parasite challenge.

## 5. Conclusions

We showed that some phenolamides are found in sunflower floral resources (i.e., nectar and pollen) while they are absent from sunflower leaves and corolla and are therefore collected by bumble bees when foraging. Both sunflower pollen and its phenolamides had detrimental effects on *Bombus terrestris*, but phenolamides had milder effects. Conversely, sunflower pollen and its phenolamides had surprising diverging effects on *Crithidia bombi* load. On the one hand, sunflower pollen did not alter parasite load, which contrasts with previous studies conducted on *B. impatiens*. On the other hand, sunflower phenolamides increased parasite load, which discredits hypotheses made in these previous studies. Because in plant–bee–parasite studies control diets and biological models differ greatly, we caution against comparisons that could be drawn and encourage future research to develop a proper standardised framework.

## Figures and Tables

**Figure 1 biology-11-00545-f001:**
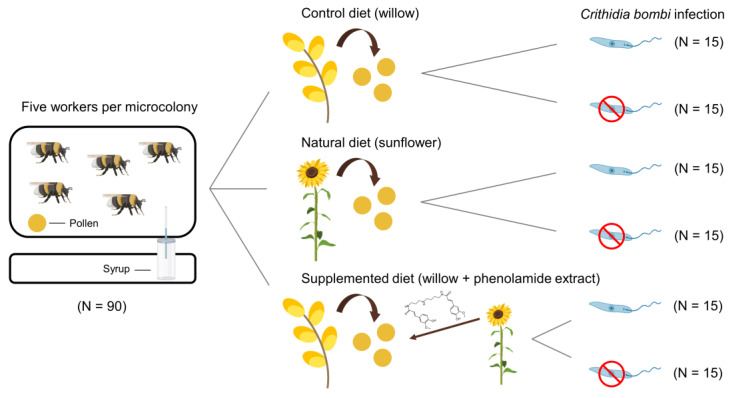
Experimental design and summary of the six treatments provided to *Bombus terrestris* microcolonies. Each microcolony consisted of five workers that were fed for 35 days. Phenolamide structure: *N,N′*-diferuloyl spermidine. This figure was created using BioRender (https://app.biorender.com/ accessed on 22 January 2022) and Flaticon (https://www.flaticon.com/ accessed on 22 January 2022).

**Figure 2 biology-11-00545-f002:**
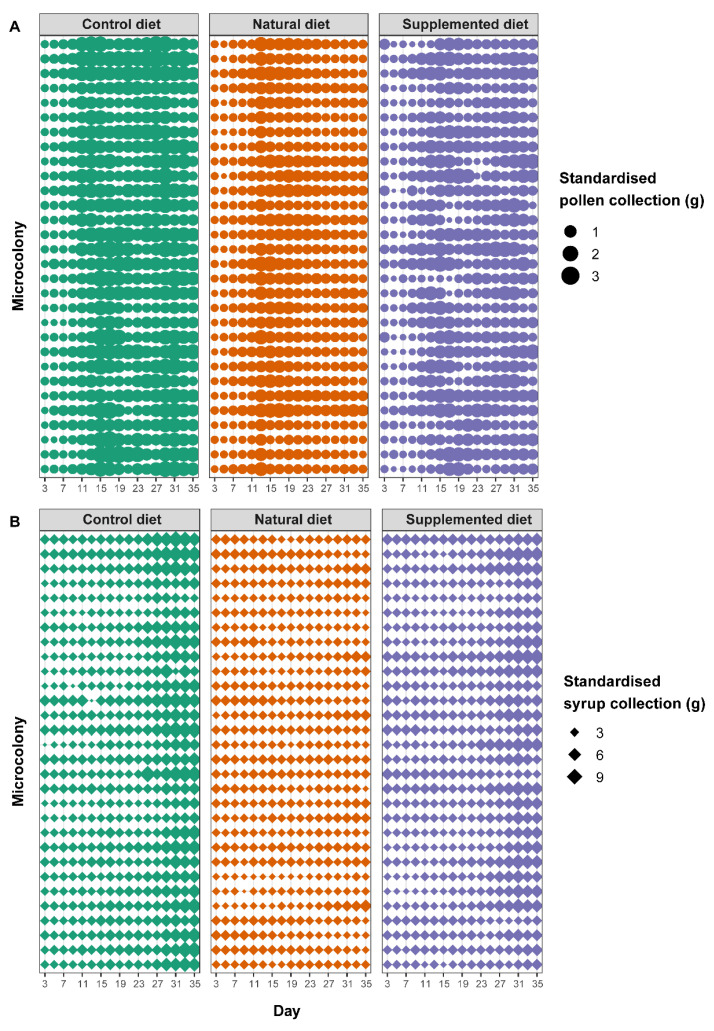
Standardised food collection in microcolonies over time. Each horizontal line represents the collection values for a single microcolony over time. The thickness of the dots or diamonds indicates the mass of collected (**A**) pollen and (**B**) syrup, respectively, per g of bee on that day.

**Figure 3 biology-11-00545-f003:**
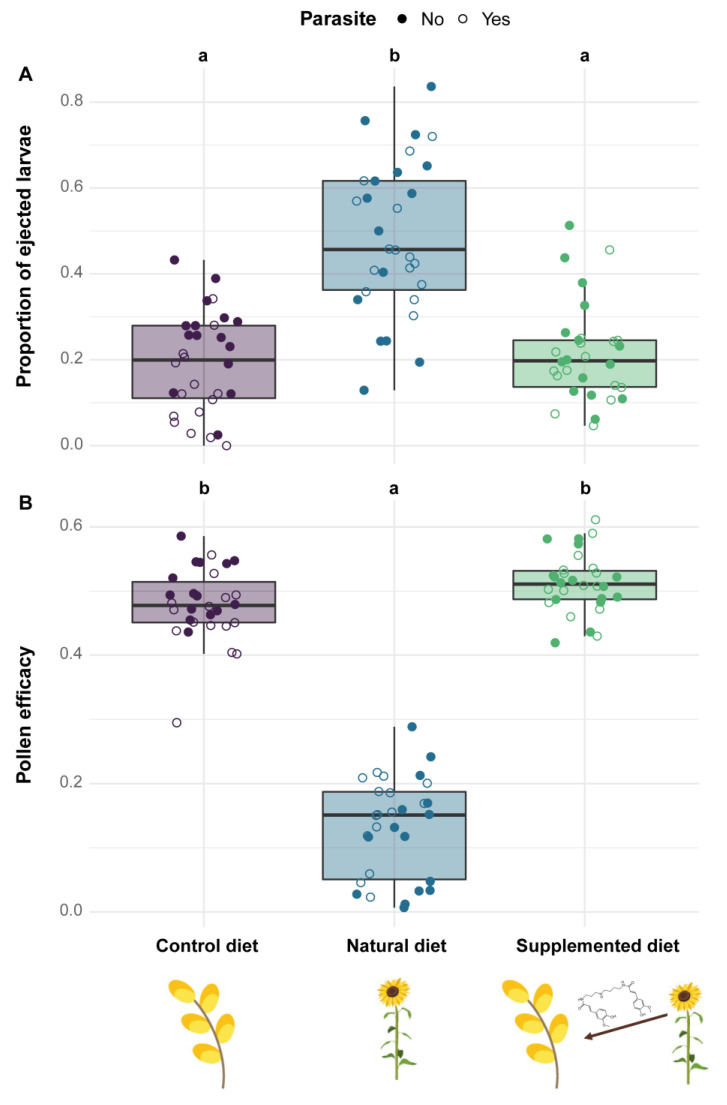
Stress responses in microcolonies across treatments. (**A**) Proportion of ejected larvae (i.e., number of ejected larvae divided by the total number of living offspring). (**B**) Pollen efficacy (i.e., the mass of hatched offspring divided by total mass of collected pollen). Two treatments sharing a letter are not significantly different (GLMMs).

**Figure 4 biology-11-00545-f004:**
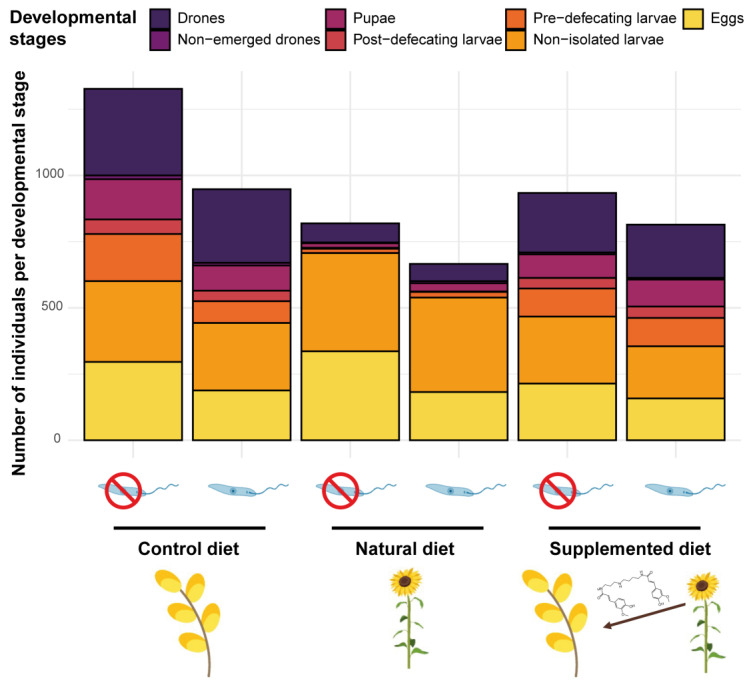
Microcolony development across treatments. Compilation of the brood composition of all microcolonies (n = 15) in the different treatments after 35 days of the experiment.

**Figure 5 biology-11-00545-f005:**
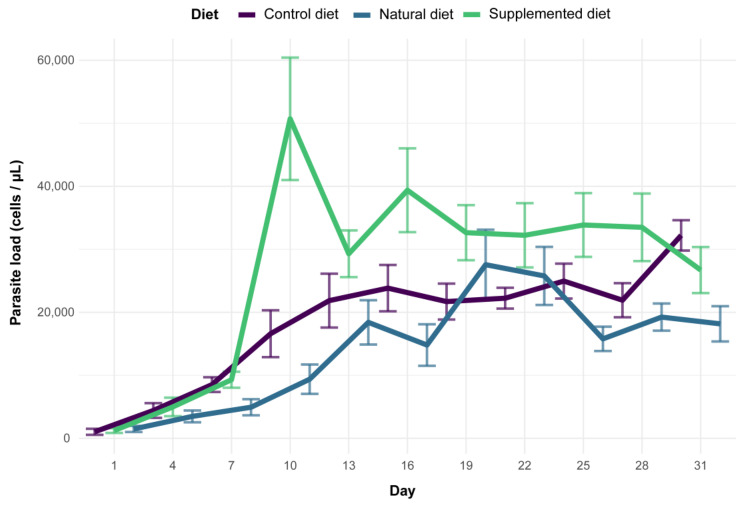
Parasite load among diet treatments for 31 days (mean ± SE). The first measure was taken three days after inoculation (day 1 in the figure) and the following measures were then taken every three days.

**Figure 6 biology-11-00545-f006:**
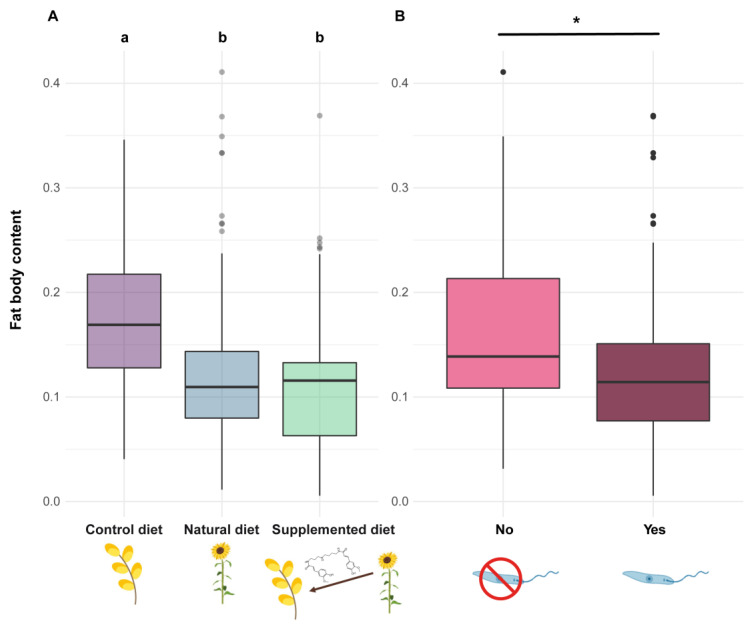
Fat body content in *Bombus terrestris* individuals. Fat body content is measured as a proportion and has therefore no unit. (**A**) Two diets sharing a letter are not significantly different (GLMMs). (**B**) The asterisk (*) indicates significant differences between parasite treatments (GLMMs).

**Figure 7 biology-11-00545-f007:**
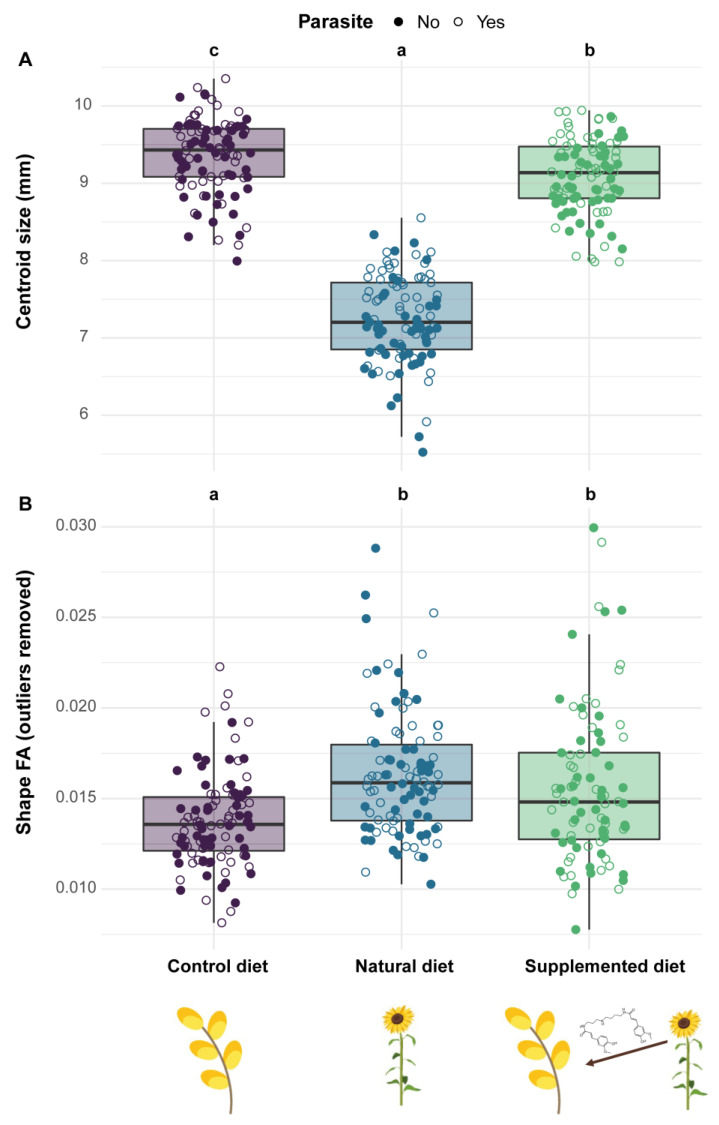
Phenotypic variation in emerged males among treatments. (**A**) The right forewing centroid size—a proxy for body size—is defined as the square root of the sum of squared distances between all landmarks and their centroid. (**B**) The shape fluctuating asymmetry (FA) is defined as the square root of the sum of each squared value of each landmark (x and y; no unit). Two treatments sharing a letter are not significantly different (GLMMs).

**Table 1 biology-11-00545-t001:** Total phenolamide content in sunflower tissues expressed as mg triferuloyl spermidine equivalent (TSE)/g tissue. Superscript letters indicate the outputs of the Kruskal–Wallis test between tissues (*p* < 0.001), with two medians sharing a letter being not significantly different. LOD: limit of detection of *N,N′,N″*-triferuloyl spermidine in a Waters™ Q-ToF US: 2.5 × 10^−6^ mg/mL.

	Leaf	Corolla	Nectar	Pollen
Plant 1	0 (<LOD)	0 (<LOD)	9.19	10.39
Plant 2	0 (<LOD)	0 (<LOD)	24.8	10.94
Plant 3	0 (<LOD)	0 (<LOD)	0.83	25.38
Plant 4	0 (<LOD)	0 (<LOD)	21.07	11.92
Plant 5	0 (<LOD)	0 (<LOD)	4.77	12.62
Mean ± SD Median (Min–Max)	0 0 ^a^	0 0 ^a^	12.12 ± 10.38 9.19 (0.83–24.8) ^b^	14.25 ± 6.18 11.92 (10.39–25.38) ^b^

## Data Availability

The authors commit to making data publicly available upon acceptance of the manuscript.

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
