# Peer review of "Poison or Potion: Effects of Sunflower Phenolamides on Bumble Bees and Their Gut Parasite"

_biology, 2022, doi:10.3390/biology11040545_

Round 1

Reviewer 1 Report

It is an interesting and meaningful study for clarifying the mechanism of natural medicinal effects in plant-bee-parasite interplays. The present study was well-designed and with a heavy workload, which provided useful information for understanding the impacts of sunflower pollen and its phenolamides on a bumble bee. 
I have a major comment for your reference only: If we want to study the medicinal effects of some plants and their metabolites on insects infected with parasite, the parasite with virulence to insect host should be adopted. However, in this study, only mild effects of the gut parasite Crithidia bombi were observed, with neither high mortality nor high stress responses. 

Author Response

Thank you for your nice comments.

We agree that addressing studies on more virulent parasites is needed in this research field.

However, we decided to focus on Crithidia bombi because of the available literature and also because it is quite easy to find it in natura. Besides using parasites with moderate virulence also allows for "long-term" studies and is more suitable for testing therapeutic effect as we did herein. A prophylactic design would be more appropriate for more virulent parasites as bumblebee individuals could die shortly after inoculation (i.e. before the possibility for therapeutic effects of pollen diets).

Reviewer 2 Report

Being to the point, I found the manuscript to be thoughtfully and well-written. As a peer reviewer, I appreciate how much care was taken to produce a near publication-ready manuscript.  I provided some minor comments for your consideration in an attached file.  Very well-done!

Reviewer 3 Report

First of all I would like to thank the authors for the excellent quality of this paper, which is clearly presented, clearly analyzed, and very well discussed, with a very good bibliographic work. I really appreciated the reading of this paper, and would only have a few minor corrections to suggest bellow.

I would suggest to have a shape filled and the other only outlined for infected vs uninfected bees, to increase readibility of the figures  7 and 3

l127 meshe's characteristics missing

 l129 when was nectar collection done during the day, it can have an impact on its composition and quantity

l221-222 please reformulate your sentence, it is clearlier in the figure 5 legend

ref 3, 4, 5, 7, 12, 48,55, 81, 82, 91, 159, 164, 165, 170,186, 187 remove capital letters not needed in the title

in the reference list, please put names of species in italic
